# Data-Driven Quantitation of Movement Abnormality after Stroke

**DOI:** 10.3390/bioengineering10060648

**Published:** 2023-05-26

**Authors:** Avinash Parnandi, Aakash Kaku, Anita Venkatesan, Natasha Pandit, Emily Fokas, Boyang Yu, Grace Kim, Dawn Nilsen, Carlos Fernandez-Granda, Heidi Schambra

**Affiliations:** 1Department of Neurology, NYU Grossman School of Medicine, New York, NY 10017, USA; avinashparnandi@gmail.com (A.P.);; 2NYU Center for Data Science, New York, NY 10011, USA; ark576@nyu.edu (A.K.);; 3Department of Occupational Therapy, NYU Steinhardt, New York, NY 10011, USA; 4Department of Rehabilitation and Regenerative Medicine, Columbia University, New York, NY 10032, USA; 5Courant Institute of Mathematical Sciences, New York, NY 10011, USA; 6Department of Rehabilitation Medicine, NYU Grossman School of Medicine, New York, NY 10017, USA

**Keywords:** stroke, motor impairment, inertial measurement unit, deep learning, out-of-distribution detection

## Abstract

Stroke commonly affects the ability of the upper extremities (UEs) to move normally. In clinical settings, identifying and measuring movement abnormality is challenging due to the imprecision and impracticality of available assessments. These challenges interfere with therapeutic tracking, communication, and treatment. We thus sought to develop an approach that blends precision and pragmatism, combining high-dimensional motion capture with out-of-distribution (OOD) detection. We used an array of wearable inertial measurement units to capture upper body motion in healthy and chronic stroke subjects performing a semi-structured, unconstrained 3D tabletop task. After data were labeled by human coders, we trained two deep learning models exclusively on healthy subject data to classify elemental movements (functional primitives). We tested these healthy subject-trained models on previously unseen healthy and stroke motion data. We found that model confidence, indexed by prediction probabilities, was generally high for healthy test data but significantly dropped when encountering OOD stroke data. Prediction probabilities worsened with more severe motor impairment categories and were directly correlated with individual impairment scores. Data inputs from the paretic UE, rather than trunk, most strongly influenced model confidence. We demonstrate for the first time that using OOD detection with high-dimensional motion data can reveal clinically meaningful movement abnormality in subjects with chronic stroke.

## 1. Introduction

Stroke commonly causes motor impairment in an upper extremity (UE), resulting in abnormal movement that interferes with function [1,2,3]. These movement abnormalities are underpinned by the loss of volitional motion, poor control of motion, and/or intrusive muscle co-activations, all of which hinder activities of daily living (ADLs) such as grooming or feeding. A major goal of stroke rehabilitation is to restore UE function by guiding abnormal movements toward more normal patterns.

Rehabilitation therapists observe the performance of ADLs to identify movement abnormalities, for which knowledge about healthy movement is needed. Sighted individuals have built an internal representation of what healthy movement looks like from decades of observing other humans move (hence the uncanny valley phenomenon [4]). Rehabilitation clinicians have sharpened this internal representation with domain training and experience, which inform their identification of abnormal movement patterns resulting from stroke and other clinical disorders. This observational approach supports the rapid appraisal of movement abnormality but is challenging to communicate in a standardized manner and is liable to be influenced by level of clinical experience. An ideal approach would provide a rapid assessment of movements in a functional context while also providing a standardized and detailed measurement of abnormality.

Clinical instruments, such as the Fugl-Meyer Assessment (FMA) [5], Action Research Arm Test (ARAT) [6], and Wolf Motor Function Test (WMFT) [7], have good standardization and solid clinimetric properties [8] but are limited by imprecision and impracticality. Their ordinal scales coarsely score the quality of movements [9,10] and their administration requires additional time (30–45 min), a trained assessor, and specialized testing kits. These limitations reduce their utility in clinical and research settings to provide granular and frequent feedback about movement abnormality.

Optical- and sensor-based motion capture systems generate quantitative metrics of movement and can more precisely detail abnormality [11,12,13]. Most commonly, these kinematic setups evaluate simplified planar motions to virtual targets while arm weight is supported and joints are locked out [14,15,16]. The kinematic patterns of these constrained motions differ from real-world functional movements, where a stroke patient engages an actual object while managing weakness, gravity, and multi-joint control [12,17,18,19,20]. Unconstrained 3D motion capture allows better replication of real-world conditions and thus provides more functionally relevant read-outs [12,21,22,23,24,25,26]. However, many kinematic variables are produced with higher-dimensional motion capture, leading to a computational bottleneck. Investigators have typically relied on variable preselection, feature engineering, or dimensionality reduction to manage computational complexity, but these steps risk human selection bias or the omission of relevant information [11,27].

Combining motion capture with deep learning (DL) may provide a solution for appraising the complex, nonlinear, multivariable data generated by functional movements. DL models automatically learn data features and relationships that are most informative for inference, thus avoiding feature selection bias or information loss. An emerging strategy, called out-of-distribution (OOD) detection [28,29,30], could be used to measure movement abnormality in complex kinematic datasets. In this approach, a predictive classifier first learns the data features (patterns) of a training dataset and is then tested on a previously unseen test dataset. If data features of this test set are shifted, i.e., OOD from the training set, the DL model is less confident in its classifications and assigns them lower prediction probabilities [28,30]. If the test set consists of stroke data, lower prediction probabilities signal an OOD deviation from the learned normal data features. In other words, lower prediction probabilities flag abnormal data.

OOD detection has been increasingly used in medical image processing, for example, to identify and score brain lesions in MRIs [31], to distinguish malignancies in histopathological samples [32], to detect skin cancer in images [33], and to detect malaria in blood smears [33]. OOD detection has also been applied to motion data captured by sensors [34] or video [35] to detect new activities unrelated to trained activities. To our knowledge, this strategy has not been applied to stroke motion data to detect abnormal movement patterns. Here, we sought to identify movement abnormality in stroke subjects using sensor-based motion capture and OOD detection. We trained two DL models on healthy subjects performing a 3D functional activity to learn the data patterns of functional primitives (reaches, transports, repositions, stabilizations, idles [36]). We anticipated that motor impairment after stroke would shift the data distributions of these primitives, resulting in diminished model confidence and lower prediction probabilities.

## 2. Materials and Methods

### 2.1. Subjects

We studied 29 healthy subjects and 50 chronic stroke subjects (Table 1). Enrolled subjects were ≥18 years old, right-hand dominant (premorbid in stroke), and able to comprehend language and the tasks involved. Stroke subjects had a unilateral ischemic or hemorrhagic stroke at least 3 months before resulting in contralateral UE weakness (Medical Research Council score < 5/5 in any major muscle group). We excluded individuals with subarachnoid or intraventricular hemorrhage; hemorrhagic stroke with mass effect; traumatic brain injury; musculoskeletal, major medical, or non-stroke neurological condition that interferes with motor function; contracture at shoulder, elbow, or wrist; moderate UE dysmetria or truncal ataxia; visuospatial neglect; apraxia; global inattention; or legal blindness. Stroke was confirmed with the radiographic report. Lesions in non-motor areas and in the opposite hemisphere were permitted barring bilateral weakness. Both ischemic and hemorrhagic stroke were included, as chronic motor deficits do not substantially differ between the two types [37].

All subjects gave written informed consent in accordance with the Declaration of Helsinki. The study was conducted with Institutional Review Board approval at the New York University Grossman School of Medicine. We previously reported the development of a primitive-counting pipeline that used some of the stroke subjects and models examined here [38,39].

### 2.2. Functional Activity

Subjects performed a semi-structured functional activity entailing the movement of a hand-sized object around a horizontal target array (Figure 1). Movements were made freely in three dimensions; no trunk restraint, joint braces, or body weight supports were used. Subjects were seated at a table with the target array printed on a mat. The array consisted of a center target and eight outer targets (target diameters 5 cm; array diameter 48.5 cm). Subjects were asked to extend their arm (nonparetic side for stroke) to the furthest target while we observed their trunk. The mat was positioned so that this distant target would not elicit trunk flexion.

Subjects were instructed to move a toilet paper roll between the center and each outer target at their natural pace, returning the UE to rest after each movement. Skillfulness (i.e., accuracy and speed) was neither emphasized nor measured. Healthy subjects performed five trials of the target array with each UE (10 trials total), whereas the stroke subjects performed five trials with their paretic UE only.

The tabletop activity generated multiple samples of functional primitives, elemental movements that are strung together to complete an ADL [36]. We previously identified five main classes of functional primitives: reaches (UE movement into contact with a target object), transports (UE movement to convey a target object), repositions (UE movement proximate to or away from a target object), stabilizations (minimal UE motion to keep a target object still), and idles (minimal UE motion to stand at the ready near a target object). Three primitive classes are motion-based (reaches, transports, repositions) and two have minimal or no motion (stabilizations, idles). Given the task requirements, the tabletop activity mainly generated motion-based primitives.

A trained assessor administered the UE FMA to all stroke subjects, which produced the ground truth estimate of clinically relevant movement abnormality. The UE FMA consists of 33 scored movement items, with the maximum total score of 66 indicating normalcy. Of note, the FMA rates UE movements not only by whether they can be completed in their entirety but also by the intrusion of abnormal flexor and extensor synergy patterns [5].

### 2.3. Motion Capture

During the functional activity, we recorded upper body motion with nine inertial measurement units (IMUs; Noraxon, Scottsdale, AZ, USA). IMUs were affixed to the bilateral hands, forearms, and arms, and the C7, T10, and sacral spine (Figure 1). Sensors were approximately positioned relative to bony landmarks (e.g., the forearm sensor at three finger-breadths proximal to the wrist crease, the arm sensor midway between elbow and shoulder, etc.). By introducing some variation in the sensor data based on location, this step built in robustness to minor differences in sensor placement across individuals.

Each sensor measured 3D angular velocity, 3D linear acceleration, and magnetic heading at 100 Hz. Motion capture software (myomotion, Noraxon) used the angular velocities to generate 3D unit quaternions per sensor, which we transformed to a sensor-centric framework using coordinate transformation matrices. The software also generated 22 anatomical angles for the upper body (Table 2) by applying a proprietary height-scaled skeletal model to the IMU data. The sensor array thus generated a high-dimensional kinematic dataset with 76 dimensions (9 IMUs × 3D accelerations, 9 IMUs × 3D quaternions, 22 joint angles) every 10 ms.

To generate ground truth labels for the IMU data, we used video synchronized with the IMU recordings. Upper body motion was recorded at 60 FPS with two high-definition cameras (1088 × 704 resolution) placed orthogonally ~2 m from the subjects. Human annotators were trained on our taxonomy of functional motion [36] to identify the visual appearance and boundaries of primitives. Once they achieved high interrater reliability with our experts (AP, EF; Cohen’s K ≥ 0.96), the annotators segmented the beginning and end of each primitive on the video recordings. This applied ground truth labels to the corresponding IMU data. We used these labeled IMU data for model training and testing.

### 2.4. Primitive Classification

To enable the DL models to learn the data patterns that are in-distribution with healthy functional movements, we trained the models to learn the motion features that identify and distinguish primitive classes exclusively on healthy patients. We speculated that these motion features would also contain information about movement quality. Having the models focus on the motion features that distinguish primitives forces them to ignore information that is irrelevant to abnormality detection, such as idiosyncratic motions of individuals, target order, resting posture, or physical characteristics, such as arm length.

### 2.5. Model Selection

Because we were interested in how well OOD detection could work as a general approach for identifying movement abnormality, we trialed separate DL models with distinct architectures: a sequence-to-sequence (Seq2Seq) model and action segment refinement framework (ASRF). We previously showed that these models could effectively learn primitive classification from IMU and discuss their development and architectures in detail elsewhere [38,39,40].

Briefly, our approach involves using six-second data windows as inputs for our models. Within each window, the models focus on predicting primitives within the middle 4 s. The remaining 1-s segments on either side of the middle segment provide valuable temporal context for the prediction process. It is important to note that the six-second data windows do not correspond directly to segmented primitives. Instead, they serve as an input for predicting the primitives within the middle 4-s segment. It is possible for this segment to contain multiple primitives, each primitive with a duration typically ranging from 0.5 to 1 s. The Seq2Seq model aims to determine the correct order of the primitives within the window, rather than identifying the exact time step of each primitive.

The Seq2Seq model consists of a three-layer bi-directional gated recurrent unit (GRU) encoder and a single-layer GRU decoder. Seq2Seq encodes the sequence of primitives from the input motion data into a hidden vector, which is then decoded into a predicted sequence of primitive classes performed by the subject.

The ASRF consists of two convolutional neural network modules: a segmentation module to perform pointwise predictions of the primitives and a boundary detection module to demarcate primitive boundaries. The output of the segmentation module is refined using the boundaries detected by the boundary detection module. Between two detected boundaries, one primitive is selected based on the majority vote. Once the segmentation output is refined, the resultant output is the predicted sequence of primitive classes performed by the subject.

For each model, its final softmax layer assigns normalized probabilities to each primitive class. Both models use a winner-take-all approach for classification, selecting the primitive class with the maximum softmax probability as the final prediction. Hence, for each predicted sequence of primitives, we used the winning probability scores to examine model uncertainty (see Section 2.8, below).

### 2.6. Model Training and Testing

We trained Seq2Seq and ASRF on the healthy subject data using a leave-one-subject-out approach (Figure 2). We iteratively held out one healthy subject from each training run, using the remaining healthy subjects for training (*n* = 24) and validation (*n* = 4). We performed this iterative training to enable the models to learn the motion features of all healthy subjects, which avoided training biases introduced by a single training subset.

The training runs produced 29 healthy subject-trained versions of the Seq2Seq and ASRF models (which we henceforth refer to as “healthy-trained”). We used these healthy-trained models to classify primitives in the healthy test subject who was held out during training (*n* = 29). This allowed us to determine the performance and confidence of a healthy-trained model on new healthy data, which benchmarked model confidence when classifying in-distribution data. We then used the 29 healthy-trained versions of each model to classify primitives in stroke subjects (*n* = 50).

### 2.7. Assessment of Classification Performance

To first identify classification performance by both models, we compared their predicted primitive sequences against the ground truth sequences using the Levenshtein sequence-comparison algorithm [38,39,40,41] (Figure 3). This algorithm identifies the changes needed to convert the predicted sequence to the ground truth sequence. We counted the number of required deletions (removing a primitive when the model incorrectly added it), insertions (adding a primitive when the model incorrectly missed it), and substitutions (swapping in a corrected primitive when it was misclassified).

From these counts, we computed the false discovery rate (FDR) and true positive rate (TPR) as follows:$$\mathrm{FDR}=\frac{\left(\#\;\mathrm{deletions}+\#\;\mathrm{substitutions}\right)}{\#\;\mathrm{predicted}\;\mathrm{primitives}}$$
$$\mathrm{TPR}=1-\frac{\left(\#\;\mathrm{insertions}+\#\;\mathrm{substitutions}\right)}{\#\;\mathrm{ground}\;\mathrm{truth}\;\mathrm{primitives}}$$

From these metrics, we calculated the F_1_ score to assess model classification performance on each test set. The F_1_ score represents the harmonic mean between TPR and FDR and ranges between 0 and 1; a value of 1 indicates perfect classification performance. It is calculated as:$$F_{1}=\frac{2\;\mathrm{TPR}\;\left(1-\mathrm{FDR}\right)}{\mathrm{TPR}+\left(1-\mathrm{FDR}\right)}$$

### 2.8. Assessment of Model Confidence

The central aim of our study was to determine whether the confidence of a healthy-trained model could flag abnormality in functional movements. We thus focused on the prediction probabilities assigned to the classifications of motion-based primitives—reaches, transports, and repositions—by Seq2Seq and ASRF (Figure 2). Model classification generated a cluster of ~450 prediction probabilities per healthy test subject (one model version, both UEs) and ~7500 probabilities per stroke test subject (29 model versions, one UE).

We predicted that model confidence would decrease when encountering the movements made by stroke subjects compared to healthy subjects. We pooled the prediction probabilities for the healthy and stroke test sets and used generalized linear mixed models (GLMMs) to test the fixed effect of subject group (healthy vs. stroke) on the probability score. Subjects were treated as a random effect to adjust for within-subject correlations between multiple observations.

We also predicted that model confidence would diminish with greater levels of motor impairment. We thus stratified stroke subjects by their impairment category (mild, FMA score 53–65; moderate, 26–52; severe, 0–25 [42]; Table 1) and examined their pooled prediction probabilities. We used GLMMs to test the fixed effect of impairment category (mild vs. moderate vs. severe) on prediction probabilities, with subjects again as a random effect. We also performed post hoc pairwise comparisons between impairment categories using two-tailed Student’s *t*-tests.

Finally, we predicted that model confidence for individual stroke subjects would scale with their individual degree of impairment. To support this analysis, we trialed various data summarization techniques on each subject’s probability cluster. We applied Gaussian and Cauchy fits and principal component analysis to generate the mean and standard deviation (SD), median and interquartile range (IQR), and their first principal components (PC1_mean/SD_ and PC1_median/IQR_), respectively. We also computed the Wasserstein distance between the probability clusters of a stroke subject and the healthy test set, which quantifies the minimum cost of transforming one data cluster to match the features of another [43,44].

We used Spearman’s correlation to evaluate the relationship between the summarized prediction probabilities (mean, median, PC1_mean/SD_, PC1_median/IQR_, Wasserstein distance) of each stroke subject and their FMA scores.

### 2.9. Localization of Reduced Confidence

Finally, we examined the UE segmental locations of data inputs that most strongly influenced model confidence. When UE motion is limited due to paresis, individuals with stroke commonly use compensatory motion, such as trunk flexion, to complete an activity. The magnitude of trunk flexion increases as elbow extension and shoulder flexion decrease [25,45,46,47]. Because our input data contain both UE and trunk kinematics, it is possible that model uncertainty stems either from the detection of altered or absent UE motion, from the detection of increased truncal motion, or from both.

To this end, we examined which sensors and joint angles most strongly reduced prediction probabilities for stroke subjects. To determine the features that influenced the confidence measure, we used the technique of computing the gradients of confidence with respect to the inputs [48]. Specifically, we computed the gradient of the predicted confidence score with respect to each input feature at the time of prediction using the backpropagation algorithm. The gradients were normalized per subject so that their sum added up to 1. This allowed us to identify the relative contributions from each sensor and joint angle to the overall gradient, which can be considered the relative importance of each data input to the final confidence score. This roughly localized which locations the models were most closely “looking at” to inform their classifications.

We applied this technique to all stroke subjects in the test set. We localized the source of reduced confidence by identifying which sensors and joint angles most strongly influenced the model’s prediction. Per patient, we computed the average percent contribution to the total gradient for each input feature. We then averaged these contributions of each input feature across patients. We selected the inputs with the highest proportion of the total gradient as the top sensor data and joint angles that most influenced confidence.

### 2.10. Analyses

We used Python 3.8 (Pytorch [49], Matplotlib [50], Numpy [51], Pandas [52], Pickle [53], and SciPy [54] libraries) to train and test the DL models. We used Matlab to transform the quaternions to a sensor-centric framework. We used Python (NumPy [51], SciPy [54], Pandas [52], and scikit-learn [55] libraries) to generate the summary statistics. We performed statistical analysis in JMP Pro 16 (SAS Institute Inc., Cary, NC, USA). Subject group and impairment category data are reported as adjusted mean ± standard error of the mean (SEM). Significance was set at α = 0.05 and was Bonferroni-corrected for multiple comparisons.

## 3. Results

### 3.1. Model Accuracy

We first confirmed that the Seq2Seq and ASRF models learned the healthy motion features that characterize primitives (Table 3). We anticipated that healthy-trained models would generalize better to healthy test data than to stroke test data, resulting in better classification performance. Indeed, for both Seq2Seq and ASRF, F_1_ scores were higher for healthy subjects than stroke subjects. The high F_1_ scores in healthy held-out test subjects indicate that both models successfully learned the motion features that distinguish primitive classes. The models also generalized reasonably well to stroke data, but their diminished classification performance indicates that stroke motion features are OOD. We next examined this OOD-ness in greater depth using model prediction probabilities.

### 3.2. Model Confidence in Healthy and Stroke Groups

We first examined whether subject demographics influenced the confidence of either the Seq2Seq or ASRF model. For the combined healthy and stroke groups, we found no significant effects of subject sex (both models: t_77_ < 1.1, *p* > 0.290), age (t_77_ < 0.47, *p* > 0.630), or side performing the activity (t_105_ < 1.1, *p* > 0.290). For stroke subjects, we also found no significant effects of years since stroke (t_48_ < 0.69, *p* > 0.490) or paretic side (t_48_ < 0.84, *p* > 0.400). These results collectively indicate that model predictions are robust to demographic characteristics that could influence motion patterns.

We examined whether model confidence differs for stroke subjects compared to healthy subjects. Seq2Seq generated significantly higher prediction probabilities for healthy subjects (0.817 ± 0.011) than stroke subjects (0.766 ± 0.008; t_77_ = 6.6, *p* < 0.001). Similarly, ASRF generated significantly higher prediction probabilities for healthy subjects (0.926 ± 0.016) than stroke subjects (0.879 ± 0.012; t_77_ = 2.3, *p* = 0.025). These results indicate that model confidence is able to signal differences in the subject type performing functional movements.

### 3.3. Model Confidence in Categories of Stroke Impairment

In stroke subjects, we next examined whether model confidence differs by motor impairment category (Figure 4). Seq2Seq generated prediction probabilities that significantly differed by impairment category (mean ± SEM: mild impairment, 0.808 ± 0.011; moderate, 0.767 ± 0.010; severe, 0.659 ± 0.017; F_2,46_ = 46.5, *p* < 0.0001). Significantly lower probabilities were generated for lower impairment categories (mild vs. moderate, t_47_ = 2.7, *p* = 0.018; mild vs. severe, t_47_ = 7.3, *p* < 0.0001; moderate vs. severe, t_47_ = 5.4, *p* < 0.0001).

Similarly, ASRF generated prediction probabilities that significantly differed by impairment category (mild, 0.930 ± 0.015; moderate, 0.894 ± 0.015; severe, 0.714 ± 0.024; F_2,47_ = 28.6, *p* < 0.0001). Significantly lower probabilities were generated for the severely impaired category (mild vs. severe, t_47_ = 7.4, *p* < 0.0001; moderate vs. severe, t_47_ = 6.3, *p* < 0.0001) but were comparable for the mild and moderate impairment categories (mild vs. moderate, t_47_ = 1.7, *p* = 0.208). These results show that model confidence, on average, can signal differences in the impairment category of stroke subjects performing functional movements, and that Seq2Seq marginally outperforms ASRF in this capacity.

### 3.4. Model Confidence in Individual Stroke Impairment

We further examined whether model confidence relates to movement abnormality in individual stroke subjects. Regardless of how we summarized prediction probabilities, model confidence significantly correlated with individual impairment level (Table 4).

For both Seq2Seq and ASRF, the median prediction probabilities and Cauchy PC1_median/IQR_ showed the strongest associations with FMA (median probabilities shown in Figure 5). These results show that model confidence can signal the individual level of movement abnormality of stroke subjects performing functional movements.

### 3.5. Locations Driving Model Uncertainty

Finally, we analyzed the gradient magnitudes of the network output with respect to its inputs in healthy subjects and stroke patients. This analysis allowed us to identify the locations of data inputs that most strongly influence model confidence. In the healthy held-out data, the top features that contributed to both models’ confidence were shoulder flexion, elbow flexion, arm acceleration (x-, y-, z-directions), and hand rotation (y-direction) in the moving UE. This result indicates that, during training, the models learned to focus on the moving UE to classify primitives.

In stroke patients, the focus remained on the (moving) paretic UE (Figure 6, only ASRF shown for parsimony). For Seq2Seq, the top features that contributed most to model confidence were paretic UE shoulder flexion and abduction, elbow flexion, arm acceleration (x-, y-, z-directions), and forearm acceleration (z-direction). For ASRF, top features were paretic UE shoulder flexion and abduction, elbow flexion, arm acceleration (x-, y-, z-directions), and arm rotation (y-direction). For both models, thoracic/pelvic motion and nonparetic UE motion were less influential.

These results are consistent with our training approach, where the healthy training subjects kept their trunk and unused UE stationary during task performance. Readings from these non-moving joint angles and sensors are not informative for discriminating between different primitives. As a result, the models learned to rely primarily on motion features from the moving UE to discriminate between different primitives.

When applied to stroke subjects, the models continued to rely on features from the moving UE to perform classification. When paretic UE motion is altered or diminished, this reliance leads to reduced confidence and lower prediction probabilities. Importantly, truncal or nonparetic UE motion did not emerge as a major source of model confidence in stroke patients. This finding indicates that model confidence reads out abnormalities in paretic UE movement, rather than the degree of compensatory motion.

## 4. Discussion

Current approaches for measuring UE movement abnormality after stroke are hampered by imprecision, impracticality, or disconnection from a functional context. These limitations impede the assessment of recovery and the generation of feedback to guide rehabilitation training. We thus sought to develop an approach to pragmatically quantitate movement abnormality during performance of a functional activity. Our approach combined sensor-based motion capture, deep learning classification of functional primitives, and out-of-distribution detection. In DL models trained on healthy motion, we found that prediction probabilities flagged a distribution shift in stroke data. Prediction probabilities were lower in stroke subjects than in healthy test subjects, scaled with stroke impairment category, and correlated with impairment level in individual stroke subjects. We also showed that the informative inputs for the models were shoulder and elbow flexion, arm and hand acceleration, and shoulder abduction. These findings provide proof of principle that OOD detection can be used as a strategy to generate objective, quantitative information about movement abnormality from high-dimensional kinematic data.

We used two DL models, Seq2Seq and ASRF, to learn the motion patterns of functional primitives from a labeled healthy training set. We purposefully undertook this supervised training to focus model attention on motion features of primitives and to ignore non-functional or idiosyncratic information. The trained models delivered high F1 scores in the healthy test set, confirming that they successfully learned the motion features that distinguish primitive classes. As expected, the models performed less robustly in stroke data.

In theory, a healthy-trained model applied to stroke data is likely to have poorer classification performance, from which we could infer a distribution shift. In reality, labeled stroke datasets are scarce, so estimating classification performance (which requires ground truth labels) is challenging. We thus focused on prediction probabilities from the healthy-trained models, which are always generated during inference, to identify distribution shifts. We found that prediction probabilities from the healthy-trained models dropped significantly for stroke motion data, confirming a distribution shift in stroke motion features relative to healthy motion.

We next examined whether this distribution shift reflected clinically relevant motor impairment, using UE FMA as the gold-standard comparison. We found that prediction probabilities diminished with greater UE impairment, both by impairment category and by individual impairment score. We furthermore confirmed that model predictions are influenced by abnormal arm and shoulder motions in the affected UE rather than increased compensatory motions in the trunk [45,46]. These results indicate that the motion features that are used by the models to identify functional primitives also contain clinically relevant information about movement abnormality.

These significant associations demonstrate concurrent validity with the UE FMA; our approach similarly evaluates and reports on abnormal movement. Whether the detected movement abnormality is based on diminished motion or the intrusion of flexor/extensor synergies is less clear. We note that the FMA also does not make this distinction.

### 4.1. Previous Work

Our OOD detection approach with DL models advances existing strategies to measure movement abnormality from detailed kinematics. Most simply, healthy and stroke kinematic data can be compared using inferential statistics [11], but this analysis requires the pre-selection of specific kinematic variables (e.g., velocity, endpoint accuracy, movement duration), limited recording locations (e.g., wrist, shoulder, hand), and/or manual feature engineering (e.g., mean, max, entropy). Moreover, optimal kinematic variables for measuring abnormality have not been established and feature engineering may overlook relevant information [11,27].

Principal component analysis (PCA) is an alternative analytical approach that incorporates kinematic data from multiple sources (i.e., body locations, data types) into lower-dimensional representations [56]. These principal components capture most of the variance of the motion while retaining individual differences. Once derived, principal components can be compared to healthy or expert populations to measure abnormality or skillfulness, for example in planar reaching by stroke subjects [14,15] or in skiing runs by non-experts [57]. Principal components can also serve as inputs for classifiers, for example to identify subjects’ sex or joint disease from ambulation patterns [58,59] or athletic ability from whole-body motion [60,61]. The advantage of PCA is that its inclusion of several kinematic variables allows a more comprehensive and unbiased evaluation of motion. One disadvantage is that standard PCA cannot computationally handle high-dimensional, nonlinear datasets such as our motion dataset. This can result in the loss of important information and may lead to inaccurate or incomplete results. Another disadvantage of PCA is that it assumes a Gaussian distribution of the underlying data, which may not always be the case in practice. A final disadvantage of PCA is that it focuses on data with maximum variance. For functional movements, these data are unlikely to reflect the subtle features that support the identification of primitives. In contrast, our approach is based on models that are explicitly trained to focus on these subtle motion features, which in turn allows the models to flag subtle motion anomalies.

To handle the higher dimensionality and nonlinearity of human motion data, investigators have turned to machine learning (ML). To date, studies have largely focused on training ML models to directly predict clinical assessment scores (e.g., FMA, WMFT) or athletic level from kinematic data. These data are often acquired from the administration of the clinical assessments themselves [62,63,64,65,66], robotic platforms [67], or movement batteries [68,69], which require expert assessors, extra time, and/or specialized setups. Feature engineering must also be undertaken to support ML-based analysis. In addition, directly predicting an impairment score using ML-based models is not optimal, because a large and diverse dataset is required to train a robust ML model. In most practical scenarios, the number of subjects available to train such a model is limited to 50–100. This can lead to model overfitting and a lack of generalizability, where the model may perform well on the training data but fail to generalize to new, unseen data.

We used DL models because they can handle the complex, nonlinear, and multivariable data generated by human UE motion. DL models automatically learn relevant features and relationships directly from the data. Hence, they potentially improve performance while reducing the domain expertise needed for data preselection, feature engineering, or dimensionality reduction. This capacity allows us to evaluate all available kinematic data without introducing human bias. Importantly, our goal is to move beyond the ordinal and relatively coarse grading systems of the FMA, WMFT, and ARAT. Prediction probabilities offer continuous read-outs, which may provide a more nuanced capture of movement abnormality.

### 4.2. Practical Considerations

We endeavored to make the measurement approach as practical as possible. Donning and calibrating the wearable sensors, performing the functional activity, and processing the data can be expected to take 20–25 min, which offers nominal time savings over administering the FMA directly (30–45 min). However, we devised this approach to piggyback on the sensor setup used for our rehabilitation dose-measurement pipeline [38]. With sensors already in place, measuring movement abnormality requires 5–10 min to administer the functional tabletop task. This task uses a simple target array, a common object (toilet paper), and non-expert administration, making its deployment feasible.

We expect that the proposed method will work best if the DL models are trained and tested on motion data from the same motion capture system. Here, our models were trained on 76 data streams generated by a commercial system. If these trained DL models were to be applied to data from a different motion capture system or to data from grossly mispositioned sensors, classification performance is likely to suffer. With diminished classification performance, model confidence is also likely to decrease. Whether decreased confidence still has sufficient range to distinguish between subject groups and impairment levels would need to be examined. However, the particular motion capture system that is used with this approach is unlikely to matter as much as having the same data streams available for training and testing.

### 4.3. Limitations and Future Directions

We note limitations in our study that offer future investigative directions. We used the functional activity to generate unconstrained, unassisted, 3D UE movements in and out of flexion synergy, which effectively samples motor performance and segmental control [70]. Although the tabletop activity is more naturalistic and functionally based than the FMA, it is still semi-structured and necessitates stand-alone administration. In the future, expanding OOD detection to less structured activities practiced during rehabilitation (e.g., feeding, grooming, cooking, etc.) would support the simultaneous measurement of movement abnormality during therapy delivery. We previously showed that the five classes of functional primitives account for all motions/minimal motions used in several rehabilitation activities [36]. This suggests that leveraging primitive classification to examine model uncertainty could work for other functional activities, although the extent to which the five classes of primitives constitute all ADLs has not yet been examined.

We also showed that model confidence was highly influenced by inputs from the paretic UE—shoulder flexion and abduction, elbow flexion, and arm acceleration—to perform the tabletop task. Although this analysis allowed us to determine the location of movement abnormality in this functional activity, these locations may not drive abnormality readings for other activities. Explainable artificial intelligence approaches that are based on backpropagation could be used to identify the input locations that drive model predictions, which may be quite different from human expectations [71]. Identifying these locations provides an opportunity to intervene on abnormalities that have not been considered for the execution of normal movement.

Finally, we demonstrated the convergent validity of prediction probabilities with the FMA but have not yet established other important clinimetric properties, namely reliability and responsiveness to clinical change. Identifying these properties will be important for the interpretation and usability of prediction probabilities as a metric of movement abnormality [11,72].

## 5. Conclusions

We present a novel approach that uses OOD detection to identify abnormal UE movement in individuals with stroke. Our approach, employing IMU-based motion capture and DL-based functional primitive classification, uses prediction probabilities to flag motion data that deviate from healthy patterns. We demonstrate that model confidence reflects clinically relevant movement abnormality in subjects with chronic stroke. With our approach, we address both the practical and functional limitations of previous impairment measurement approaches, leveraging a motion capture setup to quantity movement abnormality. In the future, measuring movement abnormality during rehabilitation could enable clinicians to track response to therapy and adapt treatment strategy in real time.

## Figures and Tables

**Figure 1 bioengineering-10-00648-f001:**
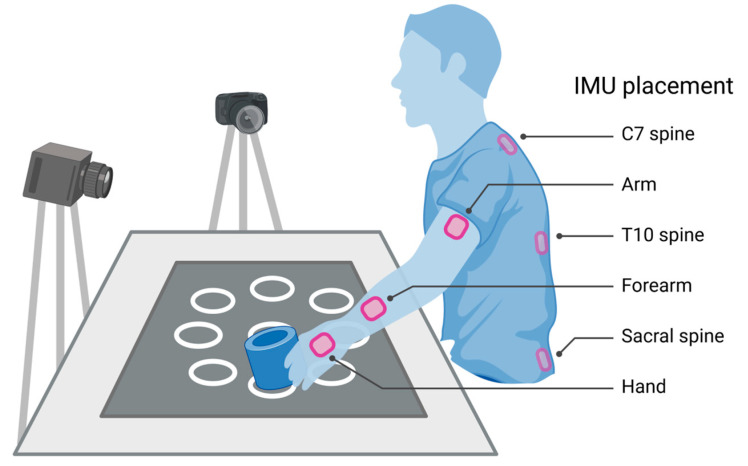
Subjects performed five trials of a semi-structured tabletop task, entailing a series of reach–transport–reposition primitives to move the object between the center target and each outer target. Nine IMUs were affixed to the upper extremities (bilateral arms, forearms, and hands) and back (C7, T10, and sacral spine); those on the back were worn under the shirt but are shown here to demonstrate location. Two orthogonal cameras captured upper body motion synchronously with the IMUs, which was used to label IMU data with their primitive classes.

**Figure 2 bioengineering-10-00648-f002:**
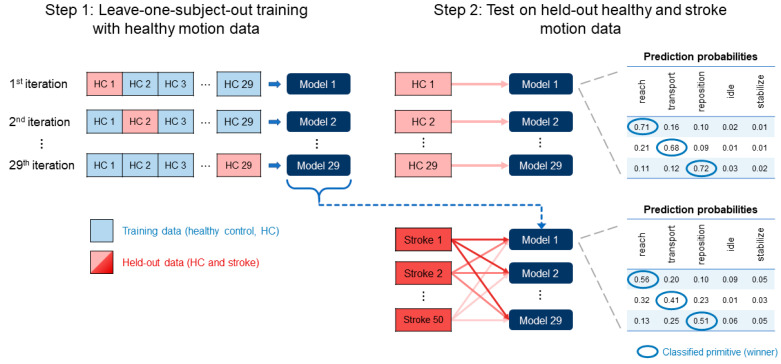
Schematic for model training and testing. This approach was separately used for the Seq2Seq and ASRF models. In Step 1, one healthy subject was iteratively held out when training each model, producing 29 healthy-trained models. All stroke subjects were held out from model training. In Step 2, the held-out subject was used to test the trained model. All stroke subjects were used to test each of the 29 healthy-trained models. Winning softmax prediction probabilities for motion-based primitives (reaches, transports, repositions) were collected from each healthy-trained model.

**Figure 3 bioengineering-10-00648-f003:**
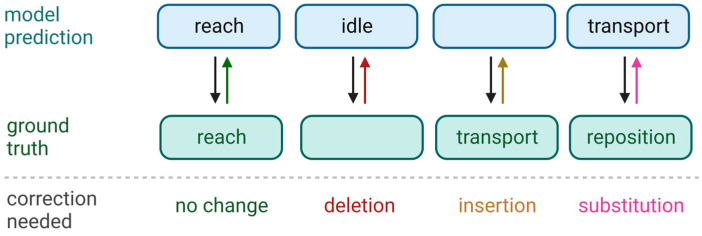
Assessment of model predictions. For both models, we used the Levenshtein sequence-comparison algorithm to identify the changes needed to convert the predicted sequence (**top row**) to the ground truth sequence (**bottom row**). Changes were counted as deletions, insertions, and substitutions.

**Figure 4 bioengineering-10-00648-f004:**
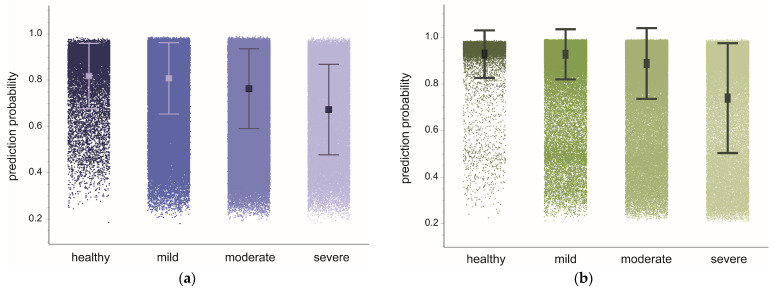
Prediction probabilities aggregated by subject group. Shown are prediction probabilities with overlaid mean and standard deviation from the healthy-trained Seq2Seq model (**a**) and ASRF (**b**). Subject groups are held-out healthy subjects and stroke subjects with mild, moderate, and severe impairment. Individual dots represent a single prediction probability for motion-based primitives. There were significant differences amongst stroke impairment categories for both models (*p* < 0.0001).

**Figure 5 bioengineering-10-00648-f005:**
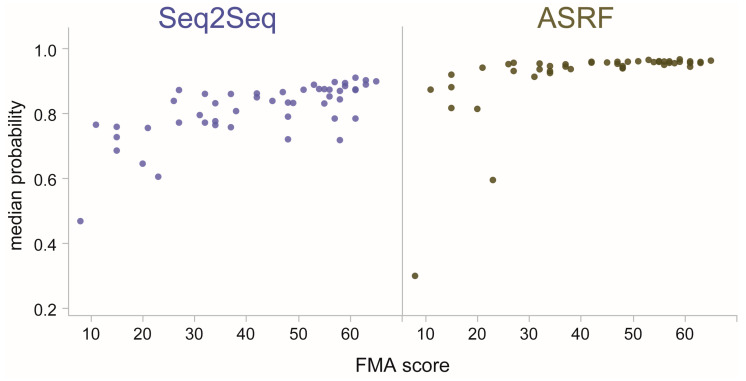
Subject-level prediction probabilities with respect to individual FMA score for Seq2Seq and ASRF. Dots are the median (Cauchy) prediction probability for motion-based primitives performed by each subject. Spearman’s correlations showed significant relationships for both models (*p* < 0.0001), indicating that higher (i.e., more normal) FMA scores related to higher model confidence.

**Figure 6 bioengineering-10-00648-f006:**
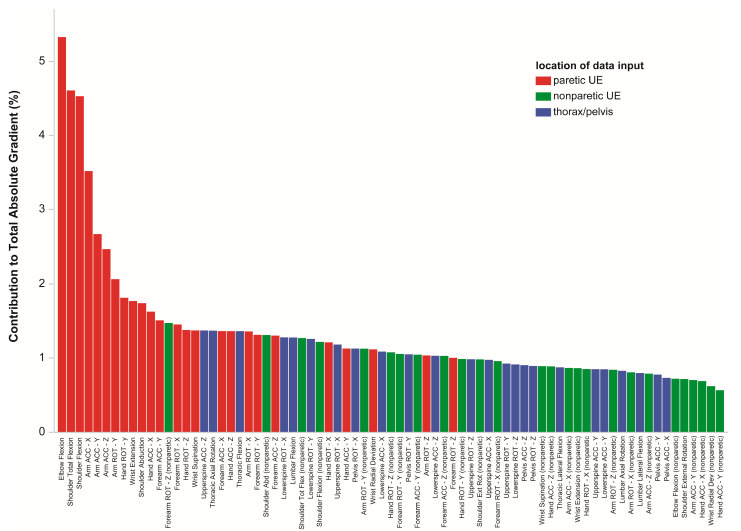
Data features influencing model confidence. Shown are data for ASRF, averaged across stroke patients. The normalized contributions to the absolute gradient (% total) from each data input indicate their relative influence on model confidence. For ASRF, the top data inputs were from paretic shoulder, elbow, and wrist angles and arm and hand acceleration/rotation. This result demonstrates that the model was focusing on motion from the paretic UE, rather than compensatory motion from the trunk or nonparetic side, to classify primitives.

**Table 1 bioengineering-10-00648-t001:** Subject demographics.

	Healthy Controls	Stroke Patients
	*n* = 29	*n* = 50
Sex ^1^	15 M, 14 F	23 M, 27 F
Age	62.4 ± 13.1 years	57.7 ± 14.0 years
Race ^2^	11 W, 14 B, 1 A, 1 AI, 2 O	23 W, 11 B, 8 A, 0 AI, 8 O
Paretic side ^3^	n/a	27 L: 23 R
Fugl-Meyer score ^4^	65.2 ± 1.0	43.1 ± 16.1
Impairment level ^5^	n/a	20 mild, 22 moderate, 8 severe
Time since stroke	n/a	5.4 ± 6.1 years

^1^ Sex: Male (M), Female (F); ^2^ Race: White (W), Black (B), Asian (A), American Indian (AI), Other (O). ^3^ Paretic side: Left (L), Right (R); ^4^ Fugl-Meyer Assessment (FMA): 0–66 points, 66 is normal; ^5^ Impairment levels: UE FMA score range 53–65 (mild), 26–52 (moderate), 0–25 (severe).

**Table 2 bioengineering-10-00648-t002:** Anatomical upper body angles.

Joint	Anatomical Angles
Shoulder	Flexion/extension, internal/external rotation, adduction/abduction, total flexion ^1^
Elbow	Flexion/extension
Wrist	Flexion/extension, pronation/supination, radial/ulnar deviation
Thorax ^2^	Flexion/extension, axial rotation, lateral flexion/extension
Lumbar ^3^	Flexion/extension, axial rotation, lateral flexion/extension

The motion capture system (myomotion, Noraxon, USA) used 9 IMUs and a proprietary height-scaled model to generate 22 upper body angles, shown in relation to their joint of origin. ^1^ Shoulder total flexion is a combination of shoulder flexion/extension and shoulder ad-/abduction. ^2^ Thoracic angles are computed between the 7th cervical (C7) and 10th thoracic (T10) vertebrae. ^3^ Lumbar angles are computed between the thoracic vertebra and pelvis.

**Table 3 bioengineering-10-00648-t003:** Classification performance of the trained models.

	Seq2Seq		ASRF	
	Healthy	Stroke	Healthy	Stroke
True positive rate	0.842	0.912	0.868	0.929
False discovery rate	0.115	0.272	0.159	0.312
F_1_ score	0.848	0.798	0.838	0.773

**Table 4 bioengineering-10-00648-t004:** Model confidence correlated with individual FMA scores.

	Seq2Seq	ASRF
Cauchy (median)	0.701 ***	0.705 ***
Gaussian (mean)	0.661 ***	0.619 ***
Cauchy PC1	−0.706 ***	−0.727 ***
Gaussian PC1	−0.660 ***	−0.600 ***
Wasserstein	−0.511 **	0.372 *

Values are Spearman′s rho. *** *p* < 0.0001, ** *p* < 0.001, * *p* < 0.01.

## Data Availability

The data are publicly available at SimTK (https://simtk.org/projects/primseq (accessed on 27 April 2023)).

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
