# Peer review of "Data-Driven Quantitation of Movement Abnormality after Stroke"

_bioengineering, 2023, doi:10.3390/bioengineering10060648_

Round 1

Reviewer 1 Report

In this paper the authors discussed the results of a study that uses deep learning algorithms to classify movement primitives in both healthy adults and stroke patients. It was found that the prediction probabilities is related to the level of motor impairment. The proposed approach depends on a simple motor task with a wearable motion sensor system. Overall, the paper is clearly written, and the results have some potential to advance clinical practice. My concerns are listed below:

1. I suggest adding a figure showing examples of classification sequence (section 2.7).

2. The segmentation needs to be clarified. It was stated that 'six-second data windows were inputted into the models'. How  are these windows related to the labeling of ground truth primitives (they may have various duration)?

3. With regards to the practical considerations, how much does the proposed method depend on the motion capture system? If one wants to use this method across clinics, there could be sensor placement differences, how will it impact the prediction probabilities?

4. The proposed method is based on a specific set of definition of 'functional primitives'. Are these primitives broad enough to generalize to different ADL movements, or new primitives need to be defined?

Author Response

In this paper the authors discussed the results of a study that uses deep learning algorithms to classify movement primitives in both healthy adults and stroke patients. It was found that the prediction probabilities is related to the level of motor impairment. The proposed approach depends on a simple motor task with a wearable motion sensor system. Overall, the paper is clearly written, and the results have some potential to advance clinical practice. My concerns are listed below:

  1. I suggest adding a figure showing examples of classification sequence (section 2.7).

We thank the reviewer for this suggestion. We have now included a schematic (Figure 3) to show how the Levenshtein algorithm compares the predicted and ground truth sequences.

  1. The segmentation needs to be clarified. It was stated that 'six-second data windows were inputted into the models'. How are these windows related to the labeling of ground truth primitives (they may have various duration)?

We have clarified this description as follows (lines 204-212):

“Briefly, our approach involves using six-second data windows as input for our models. Within each window, the models focus on predicting primitives within the middle 4 seconds. The remaining 1-second segments on either side of the middle segment provide valuable temporal context for the prediction process. It is important to note that the six-second data windows do not correspond directly to segmented primitives. Instead, they serve as an input for predicting the primitives within the middle 4-second segment. It is possible for this segment to contain multiple primitives, each primitive with a duration typically ranging from 0.5 to 1 second. The seq2seq model aims to determine the correct order of the primitives within the window, rather than identifying the exact time step of each primitive.”

  1. With regards to the practical considerations, how much does the proposed method depend on the motion capture system? If one wants to use this method across clinics, there could be sensor placement differences, how will it impact the prediction probabilities?

These are important questions which we now expand on in the Methods and Discussion, as follows:

Line 535-544: “We expect that the proposed method will work best if the DL models are trained and tested on motion data from the same motion capture system. Here our models were trained on 76 data streams generated by a commercial system. If these trained DL models are applied to data from a different motion capture system or to data from grossly malpositioned sensors, classification performance is likely to suffer. With diminished classification performance, model confidence is also likely to decrease. Whether decreased confidence still has sufficient range to distinguish between subject groups and impairment levels would need to be examined. However, the particular motion capture system that is used with this approach is unlikely to matter as much as having the same data streams available for training and testing.”

Regarding sensor placement, there should be some degree of consistency in placement location, but exact locations are not necessary. We clarify now in the methods (line 161-165):

“Sensors were approximately positioned relative to bony landmarks (e.g. the forearm sensor at three finger-breadths proximal to the wrist crease, the arm sensor midway be-tween elbow and shoulder, etc.). By introducing some variation in the sensor data based on location, this step built in robustness to minor differences in sensor placement across individuals.”

  1. The proposed method is based on a specific set of definition of 'functional primitives'. Are these primitives broad enough to generalize to different ADL movements, or new primitives need to be defined?

We expect that the proposed method could work for several other ADLs, and expand on this in the Discussion as follows (lines 553-557):

“We previously showed that the five classes of functional primitives account for all motions/minimal motions used in several rehabilitation activities [36]. This suggests that leveraging primitive classification to examine model uncertainty could work for other functional activities, although the extent to which primitives apply to all ADLs has not yet been examined.”

Reviewer 2 Report

This work demonstrates that OOD detection with high-dimensional motion data can reveal clinically meaningful movement abnormality in subjects with chronic stroke. It is meaningful and the manuscript is well-organized with detailed discuss. Thus, I suggest acceptance after minor revision.

(1) There are too many key words, please confirm which are most important and leave only 5-6 words.

(2) In fig 1, it is suggested to label the accurate position on both upper extremities and back.

Author Response

This work demonstrates that OOD detection with high-dimensional motion data can reveal clinically meaningful movement abnormality in subjects with chronic stroke. It is meaningful and the manuscript is well-organized with detailed discuss. Thus, I suggest acceptance after minor revision.

  1. There are too many key words, please confirm which are most important and leave only 5-6 words.

We thank the reviewer for this suggestion. We have now selected 5 key phrases.

  1. In fig 1, it is suggested to label the accurate position on both upper extremities and back.

We have now included labels for the sensor locations in Figure 1.